# ReaPER: Improving Sample Efficiency in Model-Based Latent Imagination

## Abstract

Deep Reinforcement Learning (DRL) can distill behavioural policies from sensory input to solve complex tasks, however, the policies tend to be task-specific and sample inefficient, requiring a large number of interactions with the environment that may be costly or impractical for many real world applications. Model-based DRL (MBRL) can allow learned behaviours and dynamics from one task to be translated to a new task in a related environment, but still suffer from low sample efficiency. In this work we introduce ReaPER, an algorithm that addresses the sample efficiency challenge in model-based DRL, and illustrate the power of the proposed solution on the DeepMind Control benchmark. Our improvements are driven by incorporating sparse, self-supervised, contrastive model representations and efficient use of past experience. We empirically analyze each novel component of ReaPER and analyze how they contribute to overall sample efficiency. We also illustrate how other standard alternatives fail to improve upon previous methods. Code for the plug-and-play tools introduced here will be made available.

## 1 Introduction

Real world agents are able to efficiently achieve goals in complex, partially-controllable environments even in situations the agent has never experienced before. This capacity can be attributed to a robust model of how the agent's actions affect their surroundings, model that is distilled from past experience, allowing the agent to estimate the effect of their actions in the pursuit of a goal in a novel situation. In practice, learning an explicit world model as in Hafner et al. (2019a); Sekar et al. (2020) is challenging in terms of sample efficiency (number of environment interactions), since the agent needs to simultaneously learn good reward-seeking behaviours and refine a generalizeable model of the environment.

This sample-efficiency challenge has been empirically observed to be more pronounced on agents that act based on sensory input (e.g., raw pixels) instead of proprioception or latent states Lake et al. (2017); Kaiser et al. (2019); Tassa et al. (2018). Since the latter is often inferrable based on raw pixels, this indicates that pixel-based pipelines have room for improvement. This is an important issue since many real-world applications are more naturally solved and articulated in terms of pixels or sensory input.

In this work we investigate several potential sources of sample-inefficiency in model-based reinforcement learning (MBRL) methods, addressing each source separately using simple techniques. We propose ReaPER (**Re**gularized **C**ontrastive Model-Based DRL with **P**rioritized **E**pisodic **R**eplay), an MBRL algorithm built upon Dreamer Hafner et al. (2019a) that addresses the identified sources of sample-inefficiency and improves results over the DMControl suite Tassa et al. (2020). We provide ablation experiments for all the introduced components, interestingly, we show that techniques that individually improve sample efficiency can sometimes show detrimental results when combined, but can be made to work together by the introduction of prioritized replay. Although current state of the art methodologies for single-task agents in the context of, for example, DMControl suite are geared towards model free learning, we specifically choose to address MBRL as a stepping stone towards continual learning in RL. An efficient Pytorch implementation of this code will be made available.

**Main Contributions.** We introduce ReaPER, an MBRL agent for control tasks in visual environments that outperforms the previous state-of-the-art MBRL agent (Hafner et al. (2019a)) on the DMControl benchmark Tassa et al. (2020). We empirically study key sources of sample inefficiency in MBRL methods, and address each individually using simple approaches, the combination of all successful approaches is integrated onto ReaPER. We experimentally show that individually promising approaches can fail to compose when combined simply, but that the introduction of prioritized replay can improve the combined approach beyond what each technique could individually achieve. This provides a self-contained ablation study of the effects of each component of the proposed ReaPER architecture.

We empirically show how ReaPER constructs a coarser but more robust world model, and how the sample efficiency improvement holds throughout model training. Ideas to improve sample efficiency that were ultimately unsuccessful such as exploration via latent disagreement Sekar et al. (2020) and bisimulation metrics Ferns et al. (2011); Ferns & Precup (2014); van der Pol et al. (2020) are also shown and discussed due to their value to improve the understanding of MBRL. Code will be made available.

## 2 Agent description

We model the environment as a partially observable Markov decision process (POMDP) with discrete time steps $t \in [1 : T]$, high-dimensional observations and rewards produced by the environment $o_t, r_t \sim p(o_t, r_t \mid o_{1:t-1}, r_{1:t-1}, a_{1:t})$, and real-valued actions selected by the agent $a_t \sim p(a_t \mid o_{1:t}, r_{1:t}, a_{1:t-1})$. The goal of the agent is to maximize the expected discounted reward $E_p(\sum_{t=1}^{\infty} \gamma^{t-1} r_t)$, with discount factor $\gamma \in (0, 1]$.

We introduce **Re**gularized **Co**ntrastive Model Based DRL with **P**rioritized **E**pisodic **R**eplay (ReaPER), a latent dynamics arquitecture extending the work of Dreamer Hafner et al. (2019a), with the following key components, which will be detailed and exploited in the coming sections:

$$
\begin{array}{ll}
\text{Sequence Augmentation} & \{o_t^q, o_t^k\}_{t=T}^{T+L} = \mathcal{SA}(\{o_t\}_{t=T}^{T+L}), \\
\text{Image encoders} & h_t^k = f_\theta(o_t^k), \; h_t^q = f_{\theta'}(o_t^q), \\
\text{Representation} & p_\theta(s_t \mid s_{t-1}, a_{t-1}, h_t^q), \\
\text{Transition} & q_\theta(s_t \mid s_{t-1}, a_{t-1}), \\
\text{Image Decoder} & q_\theta(o_t^q \mid s_t), \\
\text{Reward predictor} & q_\theta(r_t \mid s_t), \\
\text{Policy} & \pi_\phi(a_t \mid s_t), \\
\text{Value} & v_\psi(s_t).
\end{array}
\tag{1}
$$

We use the letters $p, q$ to denote true and estimated distributions respectively. To interact with the environment, observations $o_t$ are perturbed and then passed through the image encoder $h_t = f_\theta(o_t)$, the current model state is sampled from the representation $p_\theta(s_t \mid s_{t-1}, a_{t-1}, h_t)$, and the action is selected from policy $a_t \sim \pi_\phi(a_t \mid s_t)$. The full algorithm is presented below, an explanation of each component and the reasoning behind its inclusion is provided in the following sections.

### 2.1 Model learning

The model is trained on batches of sequential, observable past experience $\{\{(o_t, r_t, a_t)\}_t^{t+L}\}_{b=1}^B$, where $t$ indicates time from start of episode, $L$ is sequence length, $b$ is a batch indicator, and $B$ is the total number of batches. The model's parameters, denoted by $\theta$, are trained on a variety of complementary objectives, some of these objectives are computed independently for each time sequence (i.e., one per batch element), while others include comparisons against elements in different batches. To simplify the notation, we denote each model loss by a super-index $t$ or $t, b$ to indicate if these losses are computed across time or across time and batch respectively. The losses are described next.

**Reconstruction loss.** We follow the derivation in Hafner et al. (2019a) to increase the variational lower bound of the model (ELBO, Jordan et al. (1999)). The resulting reconstruction loss is

---

**Algorithm 1** ReaPER

---

**Require:** Hyper-parameters: Seed Episodes S, Collect interval C, Batch size B, Sequence length L, Imagi-
   nation horizon H, learning rate $lr$, Contrastive and Sparsity bonuses $\lambda_C, \lambda_{\ell_1}$, Contrastive momentum $\alpha$

   Initialize Priority Episodic replay buffer D with S random episodes
   Initialize network parameters $\theta, \phi, \psi$
   **while** *training* **do**
      **for** step $= 1, \ldots, C$ **do**
                       *Model Learning*
         Draw B sequential samples $\{(o_t, r_t, a_t)\}_t^{t+L} \sim D$
         Do sequence-consistent contrastive image augmentation $\{(o_t^k, o_t^q)\}_t^{t+L} = \mathcal{SA}(\{o_t\}_t^{t+L})$
         Compute model states $s_t \sim p(s_t \mid o_t^q, a_{t-1}, s_{t-1})$
         Compute model component $\mathcal{L}_{REC}, \mathcal{L}_{\ell_1}, \mathcal{L}_C$ as in equations 2 3 4.
         Compute model loss $\mathcal{L}_M = \mathcal{L}_{REC} + \lambda_{\ell_1}\mathcal{L}_{\ell_1} + \lambda_C\mathcal{L}_C$
         Update buffer sample priorities according to $\mathcal{L}_M$
         Update model parameters on model loss $\theta \leftarrow \theta - lr\nabla_\theta(\mathcal{L}_M)$
         Update key encoder parameters $\theta' \leftarrow \alpha\theta' + (1-\alpha)\theta$
                       *Policy Learning*
         Imagine latent trajectories $\{(s_\tau, a_\tau)\}_{\tau=t}^{t+H}$ for each state $s_t$
         Compute value targets $V_\lambda(s_\tau)$ as in Eq 8
         Compute policy and value losses $\mathcal{L}_\pi; \mathcal{L}_V$ as in Eq 7
         Update $\phi \leftarrow \phi - lr\nabla_\phi(\mathcal{L}_\pi); \ \psi \leftarrow \psi - lr\nabla_\psi(\mathcal{L}_V);$
      **end for**
                       *Data Collection*
      $o_1 \leftarrow \text{env.reset}()$
      **for** $t = 1, \ldots, T$ **do**
         Sample $s_t, a_t \sim p_\theta(s_t \mid o_t, a_{t-1}, s_{t-1})\pi_\phi(a_t \mid s_t)$
         Add exploration noise $a_t \leftarrow \text{clamp}(a_t + \epsilon z_t; -1, 1), \ z_t \sim \mathcal{N}(0, 1)$
         Observe $o_t, r_t \leftarrow \text{env.step}(a_t)$
      **end for**
      Add to buffer $D \leftarrow D \oplus \{(o_t, a_t, r_t)\}_{t=1}^T$
   **end while**

---

$$
\begin{aligned}
&\mathcal{L}_O^t = \ln q_\theta(o_t \mid s_t), \ \mathcal{L}_R^t = \ln q_\theta(r_t \mid s_t), \\
&\mathcal{L}_{KL}^t = -\text{KL}(p_\theta(s_t \mid s_{t-1}, a_{t-1}, h_t) \ || \ q_\theta(s_t \mid s_{t-1}, a_{t-1})), \\
&\mathcal{L}_{REC}^t = \mathcal{L}_O^t + \mathcal{L}_R^t + \mathcal{L}_{KL}^t.
\end{aligned}
\tag{2}
$$

The reconstruction loss $\mathcal{L}_{REC}$ is intuitively interpretable, the state $s_t$ needs to encode sufficient information about the environment to reconstruct the current observation and reward $o_t, r_t$ ($\mathcal{L}_O^t, \mathcal{L}_R^t$), it also needs to be compact enough to be *predictable* given the past state and action ($\mathcal{L}_{KL}^t$). The state then should capture the minimum amount of information of the past trajectory needed to reconstruct current and future observations and rewards.

**State-space compression.** The state-space representation of the model may attempt to capture irrelevant or non-generalizeable details of the environment. These details may still be consistent across the training dataset, and thus are not eliminated by the state prediction loss $\mathcal{L}_{KL}^t$. We therefore add a sparsity prior to the model state to alleviate this issue,

$$
\mathcal{L}_{\ell_1}^t = ||s_t||_1^1; \ \mathcal{L}_{\ell_1} = \mathbb{E}_p[\sum_t \mathcal{L}_{\ell_1}^t].
\tag{3}
$$

This prior is directly applied to the state-space representations, and not to the model weights as in standard $\ell_1$ regularization.

**Robustness to visual perturbations.** Contrastive learning has proven an effective technique to learn robust representations from visual input Oord et al. (2018); He et al. (2020); Chen et al. (2020a;b); Srinivas et al. (2020). We follow Srinivas et al. (2020), where a batch of sequential observations $\{\{o_t\}_t^{t+L}\}_{b=1}^B$ drawn from the dataset are augmented independently into query and

key observations $\{\{(o_t^q, o_t^k)\}_t^{t+L}\}_{b=1}^B$ (denoted by the superscript $q^1$ and $k$ respectively), these augmented pairs are fed through their corresponding encoders $f_\theta(\cdot), f_{\theta'}(\cdot)$ to produce paired encodings $\{\{(h_t^q, h_t^k)\}_t^{t+L}\}_{b=1}^B$. The objective of the contrastive loss is to match the encoded query $h_{t,b}^q$ (where we index the query both by time and sequence indicator) to its corresponding encoded key through a bilinear log-loss classifier,

$$\mathcal{L}_C^{t,b} = \log \frac{\exp\left((h_{t,b}^q)^T W h_{t,b}^k\right)}{\sum_{t',b'} \exp\left((h_{t,b}^q)^T W h_{t',b'}^k\right)}. \tag{4}$$

Here $W \in \mathbb{R}^{H \times H}$ is a learned bi-linear matrix, $(\cdot)^T$ denotes transposition, and key parameters $\theta'$ are updated via exponential moving average $\theta' \leftarrow \alpha\theta' + (1-\alpha)\theta$.

The augmentations used to generate the key-query pairs define the invariance class we induce into our image encodings. Since ReaPER is a model-based architecture that predicts future states based on current model state, it is imperative for these augmentations to be temporally consistent, otherwise the augmentation procedure is in direct conflict to the observation reconstruction loss $\mathcal{L}_O^t$ and the prediction loss $\mathcal{L}_{KL}^t$, since it introduces an unpredictable perturbation in successive observations that makes pixel-wise prediction impossible. We follow Srinivas et al. (2020) and chose time-consistent random cropping as our augmentation class, where all observations of the same sequence are cropped using the same mask. Randomized temporally-consistent cropping has the added benefit of having a low computational budget, since it can be computed as a tensorized view of the observation matrix, and requires no extra post-processing. An example is shown on Figure 1.

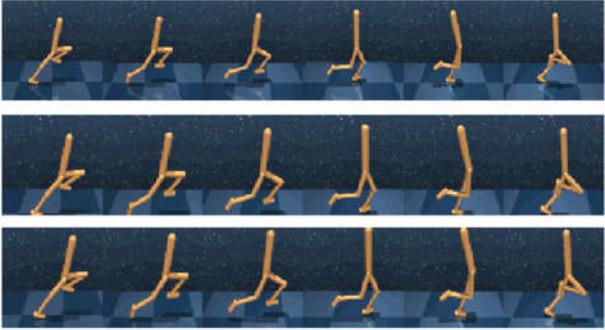

Figure 1: Random crop augmentation. Top row shows original observations from a single episode on the walker run environment, middle and bottom row show corresponding key and query observations. Note that the crop is consistent across time on both key and query. Original image is $80 \times 80$, cropped images are $64 \times 64$.

**Per-sample model loss.** The resulting per-sample model loss is a linear combination of the previous objectives,

$$\mathcal{L}_M^{t,b} = \mathcal{L}_{REC}^{t,b} + \lambda_{\ell_1} \mathcal{L}_{\ell_1}^{t,b} + \lambda_C \mathcal{L}_C^{t,b}. \tag{5}$$

Since this model loss contains both a state sparsity term and a contrastive robustness term, we expect the model to produce coarser and more robust reconstructions of its environment, this simplified and robust reconstruction yields faster policy improvement than the baseline, improving sample efficiency. Figure 2 shows a comparison between Dreamer and ReaPER reconstructions on 4 environments. In this figure we can observe that ReaPER input reconstructions have diminished fidelity on non-critical components of the image, such as the background splatter pattern, but still accurately reconstruct both the agent and the floor, the latter of which can be used as a proxy for relative motion in these environments.

---

[1]To be consistent with the literature, $q$ is used both for the estimated distributions and for the query, the context making the distinction clear.

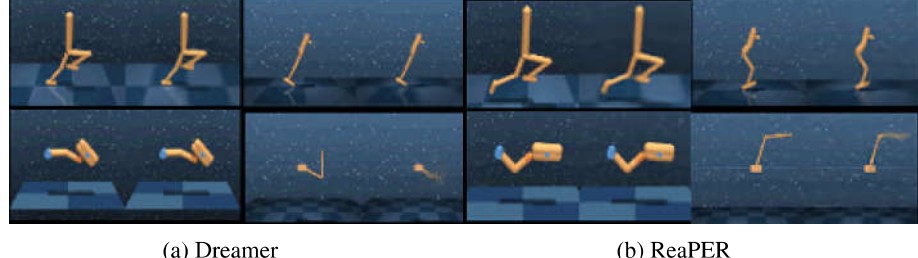

(a) Dreamer                          (b) ReaPER

Figure 2: Comparison of observation versus reconstructed observation in Dreamer vs ReaPER for the walker run, hopper hop, finger spin, and cartpole two poles environments. ReaPER builds coarse and robust reconstructions compared to Dreamer, where irrelevant background details are blurred but important elements in the environment are well modeled.

## 2.2 PRIORITIZED EPISODIC REPLAY

Prioritized experience replay Schaul et al. (2015) has been extensively used in off-policy, model-free learning (Schaul et al. (2015); Wang et al. (2016); Hessel et al. (2017), as a way to enable the learning agent to focus on high error samples from the replay buffer, and thus converge faster to a good quality agent. We extend this notion to episodic (temporally-sequential) sampling, where a sequence $\{(o_t, a_t, r_t)\}_{t=t_0}^{t_0+L}$ is sampled from the buffer with probability

$$p_{t_0} \propto \sum_{t=t_0}^{t_0+L} \mathcal{L}_M^t. \tag{6}$$

Since policy and value learning are performed exclusively on imagined trajectories, we chose model loss as our base priority measure, since experience is primarily used to train model components. Standard prioritized replay utilizes a sum-tree for efficient sampling. We extend this algorithm to temporally-sequential sampling via a tuple-valued sum-tree, where each leaf in the tree (sample) contains both its own sample loss $\mathcal{L}_M^t$ and the episodic sample loss ($\sum_{t'=t}^{t+L} \mathcal{L}_M^{t'}$); whenever a sample is fed through the model, we register $\delta_t$, the change in the sample loss, and add this sample to an update queue. Before sampling from the prioritized replay buffer, all samples in the update queue update their own $\mathcal{L}_M^t$ value and the cumulative $\sum_{t'=\tau}^{\tau+L} \mathcal{L}_M^{t'}$ values of all leaves up to distance $M$ with change $\delta_t$, the non-leaf nodes are then updated based on the cumulative loss as usual. This update is computationally efficient and only introduces a small $\mathcal{O}(L)$ linear overhead w.r.t the standard sum-tree algorithm. To our understanding, this is the first application of prioritized replay in MBRL.

## 2.3 POLICY LEARNING

**Imagined trajectories.** The latent dynamic components $q_\theta(s_t \mid s_{t-1}, a_{t-1}), q_\theta(r_t \mid s_t)$ define a fully observable Markov decision process (MDP, Sutton & Barto (2018)); we denote the time indices drawn from these imaginary trajectories by $\tau$. Imaginary trajectories start at true model states $s_t$ and then follow the MDP defined below, the policy and value functions are trained jointly on imagined trajectories on the objectives

$$\max_\phi \mathbb{E}_{q_\theta, \pi_\phi}[\sum_{\tau=1}^H V_\lambda(s_t)], \quad \min_\psi \mathbb{E}_{q_\theta, \pi_\phi}[\sum_{\tau=1}^H \frac{1}{2}||v_\psi(s_t) - V_\lambda(s_t)||_2^2], \tag{7}$$

where $V_\lambda(s_t)$ is an exponentially-weighted average of the bootstrapped value target as defined in Hafner et al. (2019a),

$$
\begin{aligned}
V_N^k(s_\tau) &= \mathbb{E}_{q_\theta, \pi_\phi}[\sum_{n=\tau}^{h-1} \gamma^{n-\tau} r_n + \gamma^{h-\tau} v_\psi(s_h), \ h = \min(\tau + k, t + H)], \\
V_\lambda^k(s_\tau) &= (1 - \lambda) \sum_{n=1}^{H-1} \lambda^{n-1} V_N^k(s_\tau) + \lambda^{H-1} V_H^k(s_\tau).
\end{aligned}
\tag{8}
$$

Note that we can differentiate through model transitions and through the reward function, so the policy can propagate gradients throughout the transitions and learn directly via gradient descent, rather than relying on policy gradients. Likewise for the value function.

## 2.4 ARCHITECTURE

We emulate the architecture of Hafner et al. (2019a), our observation encoder and decoder $f_\theta$ and $q_\theta(o_t \mid s_t)$ are implemented with a CNN and a transposed CNN respectively, the representation and transition functions $p_\theta(s_t \mid s_{t-1}, a_{t-1}, h_t^q)$, $q_\theta(s_t \mid s_{t-1}, a_{t-1})$ are jointly implemented with an RSSM Hafner et al. (2019b), which splits the state $s_t$ into a deterministic component and a stochastic component, the deterministic component is shared for the representation and transition function, and does not depend on the current image embedding $h_t$. All other components are implemented as MLPs. We note that only the contrastive component introduces an additional architectural component compared to baseline Dreamer, and thus ReaPER incurs very limited additional processing overhead compared to the former method. The experiments with each ReaPER component were designed so that minimal or no changes to the architecture were made whenever possible. For details on architecture and hyperparameters refer to section 6.1.

## 3 RELATED WORK

In this work we combine and extend ideas from diverse sources to address sample efficiency in MBRL. The works of Sekar et al. (2020); Hafner et al. (2019a) propose the use of analytic gradients through latent dynamics to train the policy, making efficient use of the world model to reduce the number of environment interactions needed to recover good behaviours from the available environment interactions.

Schaul et al. (2015) proposed the use of prioritized memory replay to enable the agent to focus on high loss samples in the experience buffer. This ensures that examples from rare or novel transitions are adequately explored by the model, reducing the need to collect a large number of transitions before these peculiarities of the environment can be learnt. While their work focused on Q-learning, here we extend their work to episodic sampling.

The use of sparse priors and data augmentation to improve performance in machine learning algorithms is well documented. Recent works in contrastive learning in particular He et al. (2020); Hénaff et al. (2019); Chen et al. (2020a) have yielded impressive results in other areas of machine learning, and recently Srinivas et al. (2020) applied these notions to augment a SAC-like agent with contrastive learning. The use of data augmentation in RL has also proven effective in Laskin et al. (2020), who also identify temporally-consistent random cropping as a particularly effective type of data augmentation for image-based pipelines in RL.

The use of alternative metrics to improve representation learning has been extensively studied Oord et al. (2018); Lee et al. (2019); Hafner et al. (2019b); one approach to avoid pixel reconstruction relies on the use of bisimulation distance Ferns et al. (2011); Ferns & Precup (2014); van der Pol et al. (2020), where two states are taken to be similar if for any action sequence effected on these states, the rewards are similar. The work of Zhang et al. (2020) addressed the use of bisimulation in DRL, and their experiments show that their agent is robust to large nuisance perturbations in the visual input. In section 6.2 we adapt this idea to latent bisimulation distance, but we observe no improvement in terms of sample-efficiency.

The work of Sekar et al. (2020) utilizes an ensemble of embedding predictors to estimate model uncertainty, this uncertainty is used as an unsupervised reward to learn a robust world model without extrinsic rewards. In section 6.3 we utilize latent disagreement to train an auxiliary exploration policy for data collection. Our results indicate that this did not improve sample efficiency in the low sample regime, further motivating our approach.

## 4 EXPERIMENTS AND RESULTS

We experimentally show the improvement in sample efficiency over a range of tasks from the Deepmind control suite. All environments feature an articulated robot that the agent controls, with contin-

uous actions in the range $[-1, 1]$, instantaneous rewards line in the range $r_t \in [0, 1]$, the maximum episode length is 1000, so all environments can accrue at most 1000 reward per episode. To isolate the contribution of each novel component of ReaPER, we first conduct an ablation study over a smaller subset of tasks. to avoid architecture changes between contrastive and non-contrastive implementations, we render observations on non-contrastive experiments at $64 \times 64$ pixels, and experiments using contrastive loss and random cropping are rendered at $80 \times 80$ pixels then cropped to ht original $64 \times 64$ resolution. The results, shown in Table 1 for the best hyperparmeters, indicate that contrastive augmentation and sparse priors can both improve sample efficiency over the benchmark, achieving higher rewards for the same number of environment interactions. Surprisingly, prioritized episodic replay by itself seemed to reduce the performance when compared against the baseline. Prioritized experience replay was, however, effective at improving sample efficiency when paired with the contrastive and sparsity auxiliary losses

Table 1: Episodic reward average for Dreamer, ReaPER, and the individual components of ReaPER (Contrastive loss, L1 and Prioritized Episodic replay, noted Contrast, L1 and PER respetively), as a function of environment steps, the conjunction of L1 regularization and Contrastive loss is also shown (L1Contrast). Rewards are averaged across the cartpole balance, cartpole swingup, reacher easy, cup catch, finger spin, walker walk, walker run, and cheetah run environments in DMControl. ReaPER consistently outperforms the other options. Prioritized episodic replay by itself does not improve upon the Dreamer baseline, but is effective in conjunction with the rest of the ReaPER pipeline. Conversely, L1Contrast under-performs both L1 and Contrast methods over all tested hyperparameters, showing that these methods do not compose well unless paired with prioritized episodic replay.

| Steps | Dreamer | Contrast | L1 | L1Contrast | PER | ReaPER |
|---|---|---|---|---|---|---|
| $100K$ | 309 | 349 | 358 | 262 | 263 | **374** |
| $200K$ | 549 | 663 | 563 | 619 | 452 | **670** |
| $300K$ | 636 | 714 | 693 | 691 | 562 | **756** |
| $400K$ | 682 | 732 | 752 | 758 | 595 | **780** |
| $500K$ | 707 | 760 | 774 | 741 | 611 | **787** |

We further validate these observations on a significantly larger set of experiments. We compare results across 18 environments and observe that, in general, ReaPER is able to obtain the same or greater rewards than Dreamer asymptotically, but it achieves these results in a smaller number of environment interactions. Figure 3a shows how the average reward over the benchmark increases as a function of environment interactions, and how this metric is improved by the addition of the regularization proposed in ReaPER. Figure 3b highlights these results by showing the average number of interactions required to achieve or surpass a given reward target for the first time over the baseline. Table 2 shows the number of steps required to reach a reward target for Dreamer and ReaPER over the same environments. We observe that ReaPER achieves better results on the benchmark across the training procedure, it also requires less samples to achieve these targets; this is especially pronounced for higher values of target rewards, since a good model is required for consistently high rewards.

Figure 4 shows the reward vs environment step curves for the best 8 environments. These environments show a marked improvement by the proposed ReaPER from the Dreamer baseline, having overall better results at each timestep. Some of the environments can continue improving with additional training time, but the results here highlight that the auxiliary objectives in ReaPER can provide a marked improvement on environments of varying complexity, making model based RL to be more effective at learning the task with less environmental samples. We also highlight that this performance increase is also marked by significantly decreased inter-seed variance.

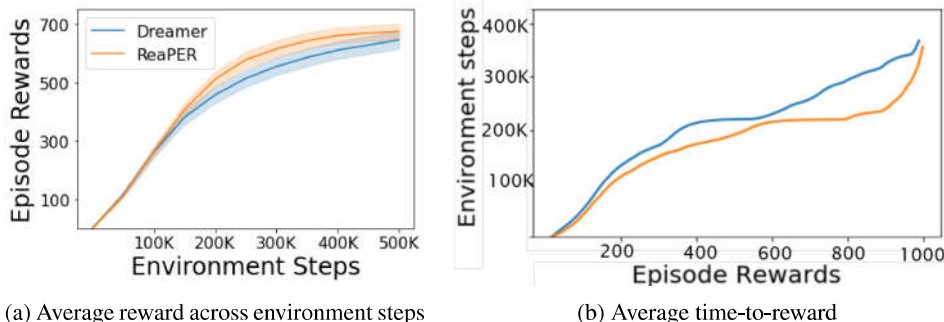

(a) Average reward across environment steps      (b) Average time-to-reward

Figure 3: Performance comparison across 18 environments of the DMControl benchmark. Left Figure shows average agent reward as a function of environment steps. Across all environments, ReaPER achieves consistently better episode rewards for the same number of environment steps. Right figure compares the average number of steps needed to first meet or exceed a reward target on the benchmark; in this metric Reaper also outperforms Dreamer across the benchmark.

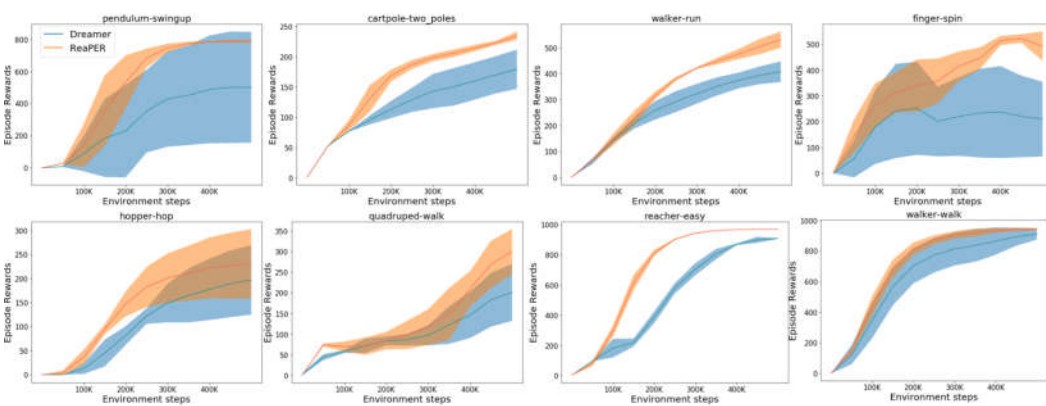

Figure 4: Performance comparison between Dreamer and the proposed ReaPER across 8 of the best performing environments in the benchmark, mean and standard deviations are computed across 3 seeds. These environments show a marked contrast in sample efficiency between the two methods, highlighting the benefit of selecting the appropriate auxiliary objectives during training.

Table 2: Time-to-reward comparison between Dreamer and ReaPER on 8 environments in DMControl benchmark. Values indicate the ratio between average number of steps required to reach reward target (expressed as percentage of maximum reward achieved by the ReaPER agent). Average number of steps computed over the seeds that reached the target, so the estimate is optimistic. Larger numbers indicate longer times for Dreamer to reach the target reward, numbers preceeded by $>$ indicate that Dreamer did not achieve the target before $500k$ timesteps.

| Reward% | Cartpole Two-poles | Reacher easy | Finger spin | Walker walk | Walker run | Pendulum swingup | Quadruped walk | Hopper hop |
|---|---|---|---|---|---|---|---|---|
| 10% | 1 | .70 | 2.21 | .87 | 2.69 | .83 | 1.10 | 1.5 |
| 30% | 1.11 | .76 | 1.68 | .90 | 1.26 | 1.69 | 1.13 | 1.98 |
| 50% | 2.47 | 1.15 | 1.35 | .95 | 1.49 | 1.75 | 1.55 | 1.37 |
| 70% | 2.15 | $>2.46$ | 1.42 | 1.01 | 1.43 | 1.84 | 1.77 | 1.64 |
| 90% | $>1.01$ | 1.64 | 1.62 | 1.49 | $>1.03$ | 1.76 | $>1.09$ | 1.81 |

## 5 DISCUSION

We study sample efficiency in the context of model based RL and propose ReaPER, an agent that learns behaviours on latent imagination and improves upon Dreamer by explicitly introducing auxiliary objectives that enable the agent to only encode and learn the parts of the environment which are relevant for behaviour learning. We ablate each novel component of ReaPER to show how each contributes to the overall agent, and experimentally validate our findings on visual control tasks in the DMControl environment suite.

Our experiments indicate that methods that have individually been shown to be effective outside of model-based RL do not necessarily translate to MBRL. Such was the case of latent disagreement for exploration and bisimulation, which have been successfully used for other purposes in RL, but proved to be ineffective at addressing sample efficiency in this context. Furthermore, components that individually do improve sample efficiency can, counterintutively, reduce performance when combined; in the case of ReaPER, this apparent anti-synergy was resolved with the addition and adaptation of prioritized episodic replay.

This paper focuses heavily on model-based architectures since we believe MBRL is better suited to handle curriculum learning and continual learning, and to environments where reward sparsity is in itself a major hurdle to overcome; a potential future direction of our research. We wish to extend this approach for other visual-based tasks that may rely on discrete control.

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
