# OpenReview forum: "ReaPER: Improving Sample Efficiency in Model-Based Latent Imagination"
_ICLR.cc/2021/Conference — Reject_

### Official Review · AnonReviewer1 · 2020-10-25

**Rating:** 4
**Confidence:** 3

**Review:**

This paper aims to improve sample-efficiency in model-based reinforcement learning (MBRL). The approach termed ReaPER is based on Dreamer (Hafner et al. 2019) and the paper integrates several ideas from prior works that are known to improve sample efficiency. Specifically, the paper borrows the contrastive learning idea from CURL (Srinivas et al. 2020) improving learning from pixels, and extends the prioritized episodic replay (PER, Schaul et al. 2015) to a more efficient version. Experiments on DeepMind control suite show that the proposed approach outperforms dreamer across 8 environments. The paper also conducts ablation study to validate the effectiveness of each design choice.


**Pros:**
++ The paper combined ideas from recent papers to improve sample efficiency of MBRL. The ablation studies shed some light on how each design helps with sample efficiency.
++ Additional experiments to show the ineffectiveness of latent disagreement for exploration and bisumulation for improving sample efficiency.
++ The overall model outperforms prior work, Dreamer.
++ The paper is relatively easy to read.

**Concerns:**
-- My biggest concern is that the paper lacks technical novelty. The overall learning framework is based on Dreamer (Hafner et al. 2019). The proposed components, contrastive learning (Srinivas et al. 2020) and prioritized episodic replay (PER, Schaul et al. 2015), are proposed by prior works and have already shown their effectiveness for improving sample efficiency in reinforcement learning. I don’t think applying these techniques to MBRL makes a compelling case.
-- The ablation studies in Table 1 seem to raise questions about the efficacy of the proposed components. For example, L1+Contrast seems to perform worse than Dreamer, L1, and Contrast. Similarly, PER alone performs the worst. It’s unclear why ReaPER, i.e., combining L1+Contrast and PER, suddenly brings the performance from the worst to the best. The magic combination that suddenly boosts the performance prevents an intuitive interpretation of the results and obscures the true effectiveness of each component.
-- I would recommend plotting the performance of each environment in the ablation study just like Fig. 7, 9, 10 in (Hafner et al. 2019). Instead of providing an average reward across different environments, providing these plots will help better understand the convergence properties of each method.


**Rating Justification:**
I commend the authors for their attempts to the important problem of sample efficiency in MBRL. However, the paper’s main issue is the lack of novelty as it is mainly combining existing techniques in very similar settings. Thus, I feel it has not met the bar of ICLR.


**Additional Comments:**
-- “Complimentary objectives” -> “Complementary objectives”
-- Missing right bracket in Eq. 8
-- I don’t think Algo 1 should occupy an entire page. The space could be saved for more detailed and self-contained explanations for Dreamer.

---

> ### Author Response · Authors · 2020-11-25
> **Response**
>
> We appreciate the constructive feedback from the reviewer; we address their main concerns below as well as in the revised paper.
>
> On Appendices 6.2 and 6.3: The methods were chosen for evaluation since they were promising ideas in the existing literature that warranted further study in the context of sample efficiency in model-based RL. We have contextualized their inclusion better in the main paper, and clarified that these were relegated to supplementary material since these methods could not improve sample efficiency in our experiments.
>
> Thank you for the additional reference, this has been added to the main paper. Code for the method proposed in the paper will be made available.

---

### Official Review · AnonReviewer2 · 2020-10-28
**Official Blind Review #2**

**Rating:** 6
**Confidence:** 4

**Review:**

This paper extends and ablates several modifications to Dreamer, the current SOTA in MBRL for control tasks. More specifically, they introduce a L1 sparsity prior on the representations, a contrastive loss with random crop data augmentation and prioritized experience replay. They also test additional changes in the Appendix, like Bisimulation distances (from Zhang et al, 2020), and exploration via latent disagreement (from Sekar et al, 2020.).

Overall, this is a well executed model exploration, and if the code gets open-sourced will be very valuable for other researchers to build upon. The results aren’t extremely strong, but do appear significant, so might still be valuable to share more broadly.

Comments/questions:
  1. Appendix 6.2 and 6.3 could use some contextualization, as they seem to come out of the blue, even though they present some valuable extra baselines. It’d be valuable to point to them from the main text as well (it is only mentioned in the Related work quite quickly).
  2. Figure 3b) should have its y-axis flipped (i..e. 0 at the bottom). It was pretty confusing to see the orange line flattening and having to interpret this as being better than the blue curve.
  3. For bisimulation, the work of van der Pol, 2020 might deserve a citation as well.

* [van der Pol, 2020], https://arxiv.org/abs/2002.11963

---

### Official Review · AnonReviewer3 · 2020-10-29
**Official Blind Review #3**

**Rating:** 5
**Confidence:** 5

**Review:**

 ##########################################################################

Summary:

This paper investigates several potential sources of sample-inefficiency and proposes an integrated algorithm to improve the sample-efficiency of model-based reinforcement learning. This paper builds upon Dreamer and incorporates sparse, self-supervised, contrastive model representations and efficient use of past experience. The experiments on DeepMind control suit show that the proposed method outperforms dreamer and the ablation study further verifies the contribution of each component of the proposed algorithm.

##########################################################################

Pros:

1. This paper tackles a valuable problem of improving the sample efficiency of model-based RL.

2. The idea of incorporating sparse, perturbation-invariant, contrastive model representations and Prioritized experience replay is interesting and promising.

3. The paper is well written and the results section is well structured. They outperform baseline methods on a popular benchmark and conduct an ablation study.

##########################################################################

Cons:

1. Novelty is limited. They use a series of techniques that have already been respectively proved to be useful.

2. This paper has a much more complex architecture than Dreamer but the performance improvement is not very significant as shown in Figure 3 but
3. The authors claim the contrastive and sparse representations are useful for sample-efficiency but they do not visualize and check the learned representations. They should first show their proposed loss functions can truly lead to more sparse and contrastive representations and then show such representations can contribute to improve the sample efficiency.

##########################################################################

Questions and suggestions:
1. In Section 2.2, why do the authors choose the model error for the sampling probability. Can you use the policy error or value error？

2. As shown in Table 1, contrastive learning and L1 regularization can individually improve the performance, but why contrastive + L1 has a negative result? Can you give some insights?

3. For Table 2, plots of learning rate will be more intuitive.

4. How to determine the weighting factor in Equation 5?


 ##########################################################################

Minor comments:
1. There are too much white space on page 3.
2. In Figure 2, it is better to compare ReaPER with Dreamer on the same state.
3. In Table 1, what does “PER” mean?

---

> ### Author Response · Authors · 2020-11-25
> **Response**
>
> We appreciate the constructive feedback from the reviewer; we address their main concerns below as well as in the revised paper.
> On the complexity of the proposed architecture: The overall architecture is marginally more complex than baseline Dreamer, since it only adds an extra visual encoder (with momentum averaged parameters) and a bilinear matrix, which are the components needed for contrastive augmentation. Paired data augmentation and prioritized replay sample do add additional auxiliary components during training, but they do not significantly increase wall time for the proposed method.
> On novelty and ablation studies: Although each component individually is indeed described in prior literature, it was not immediately apparent that these ideas could be combined to improve sample efficiency in model-based RL. Indeed, both the ablation study and the exploration and bisimulation schemes shown in Supplementary material show that translation of these techniques is nontrivial. We believe that contrastive and L1 augmentation together reduce model performance without prioritized replay since they provide an alternative path for the model to do well on average (by reducing each of these objectives individually), the introduction of prioritized replay allows the model to uniformly improve on its objective across the entire collected dataset
> On the use of other sources of error for PER: It is possible to use other sources of error for prioritized sampling. Modelling error was chosen since the experience replay buffer is primarily used to train the agent’s model. The policy and value function are trained purely on imagined trajectories and depend on the model’s reward function, so it made conceptual sense to ensure that the state and reward transition functions were well estimated.
> On the weighting factors in Equation 5: We evaluated contrastive regularization parameter and sparse regularization parameters in the ranges [1, 1e-1, 1e-2], with 1e-1 being the best parameter for both regularization objectives. A full implementation of the code will be made available.
> On Table 1: PER is used to indicate a baseline Dreamer model trained with prioritized experience replay.

---

### Official Review · AnonReviewer4 · 2020-10-29
**Model-based RL method with improved performance**

**Rating:** 4
**Confidence:** 3

**Review:**

The authors propose a model-based RL method which is built upon a baseline DREAMER [1] with additional components which empirically improve performance:
* Prioritised experience replay
* Temporally-consistent data augmentation with a contrastive loss
* L1 regularisation

# High-level comment:
* The contribution of authors is a combination of all the proposed ideas into one framework. The ideas are not novel and were considered in other works, so the contribution is really is a combination of these. From the experiments proposed by the authors, it's not easy to infer the impact of each of these ideas on the performance, the ablations do not provide a clear consistent signal and lack of explanation on why it provides an improvement. On top of that, the paper lacks experimental details, such as how the hyperparameters were optimised and what ranges were considered. It makes it hard to reproduce the experiments. More importantly, the proposed method underperforms with respect to the model-free variants which use one of these ideas, therefore it is not clear how useful the method could be for the community.

# Strengths of the paper:
* Paper is relatively clearly written and high-level ideas are easy to follow
* Equations (1) are very helpful to be able to compare the differences of the model to any other baseline

# Weaknesses:
* It is hard to judge whether the proposed combination would be a method to use due to multiple reasons. First of all, the proposed ideas are not novel and were considered in [2, 3, 4] for data augmentation, in [5] for prioritised replay. More importantly, the proposed method underperforms with respect to the model-free variants such as CURL [2] and others (cited). Why would we want at all to use a complex model-based method if what we could do is to simply use the data augmentation and get much better performance? Not clear.
* It is very good that the authors provided the ablations, but I think they do not completely answer the question of impact of each of the component. For example, for data augmentation + contrastive loss, is the most of the positive impact coming from the random crops or with both crops plus contrastive loss ? In some cases, adding L1 + Contrast underperforms with respect to Dreamer, why is it the case ? Is there any other difference besides these 3 components, such as hyperprameters, architecture or/and implementation ? If yes, the authors should report the results of their method without these 3 tricks to convince reader that it matches the performance of Dreamer.
* The impact of prioritised replay is negative but when combined with other tricks, is positive. Why is it the case ? There is no explanation of this phenomenon in the paper. Maybe it would only work for control suite and not for other problems ?
* Figure 3 shows that the difference with respect to Dreamer is quite weak.
* Figure 4 shows performance on "best" environments. It would be important to see the performance on "worst" environments. How do the results look like?

# Conclusion
Given quite weak performance of the method with respect to much simpler model-free baselines and not a clear effect of added ideas without a clear explanation on why it is the case, I would recommend to reject the paper.

---

> ### Author Response · Authors · 2020-11-25
> **Response**
>
> We appreciate the constructive feedback from the reviewer; we address their main concerns below as well as in the revised paper.
>
> Hyperparameter selection and implementation: We evaluated contrastive regularization parameter and sparse regularization parameters in the ranges [1, 1e-1, 1e-2], with 1e-1 being the best parameter for both regularization objectives. A full implementation of the code will be made available. There is no other difference between Dreamer and ReaPER, architectural or otherwise, other than what is presented in the paper. To preserve the visual pipeline architecture exactly, images are rendered at 80x80 pixels and cropped to the original 64x64 when cropping is used for contrastive augmentation.
>
> Regarding ablations and the individual effects of each component: We highlight that many of the tested components, despite being previously reported in literature, do not compose as one would expect. In fact, as the reviewer points out, L1 regularization and Contrastive augmentation work well individually, but fail to work well together over the same set of hyperparameters. The reason why this occurs, and why this detrimental effect is redressed with the addition of prioritized experience replay is in itself an interesting research question.
>
> Comparison to model-free baselines: model-free and model-based methods have alternatively held SOTA results on sample efficiency over a variety of tasks. We made a conscious choice of working with model-based baselines in this work because consider model-based RL to be a more suitable stepping stone towards curriculum learning. While we agree that current SOTA methods align with simple model-free methods, we still think it is worthwhile to investigate whether current breakthroughs in model-free methods translate into model-based approaches. The results in this paper seem to indicate that that is the case, though it would seem that translating methodologies is not a one-to-one pursuit.

---

### Decision · Program_Chairs · 2021-01-07
**Final Decision**

**Decision:**

Reject

**Comment:**

This paper explores losses and other training details to produce a model-based agent for pixel-input continuous control problems.  The authors present a rainbow-like approach that combines various separate innovations into a single system.  They show an improvement over a previous baseline on this class of problem, and break down the contributions of the various components.

Though the paper was seen as clearly written, fundamentally, the reviewers did not feel they gained insight through the presentation of the experiments.  For example, one quirk brought up by multiple reviewers is that some combinations of methods show worse performance, but then adding yet another method makes things improved relative to baseline (the authors clarified that this was with the same hyperparameters).  Reviewers found this a bit confusing and insufficiently explored (i.e. was this just hyperparameter tuning or does just the right selection tricks actually need to be combined).  This confusion around method combinations is perhaps relatively minor by itself but indicative of how this paper did not build intuition for the reviewers.  Moreover, none of the reviewers were impressed by the magnitude of improvement over the baseline dreamer agent.  While it was acknowledged the the set of methods improved things, the reviewers felt that each innovation had already been independently validated as likely to improve sample efficiency, so the fact that they did so together was not especially insightful.

I'd like to clarify for the authors that I believe this work was, in many respects, technically well executed.  Ultimately, based on the reviews and my own assessment, I don't think the scope was sufficiently ambitious considering the competitiveness of this conference.  While it is useful to occasionally produce summary works which pool a set of separate innovations, such papers must be insightful to readers, aggregate a sufficiently large number of innovations, and/or show striking performance gains.  The final reviewer scores are 4, 5, 6, 4.